# Effects of Dietary Patterns during Pregnancy on Preterm Birth: A Birth Cohort Study in Shanghai

**DOI:** 10.3390/nu13072367

**Published:** 2021-07-10

**Authors:** Zhengyuan Wang, Shenglu Zhao, Xueying Cui, Qi Song, Zehuan Shi, Jin Su, Jiajie Zang

**Affiliations:** Division of Health Risk Factors Monitoring and Control, Shanghai Municipal Center for Disease Control and Prevention, Shanghai 200336, China; wangzhengyuan@scdc.sh.cn (Z.W.); zsl13585732604@163.com (S.Z.); cuixueying@scdc.sh.cn (X.C.); songqi@scdc.sh.cn (Q.S.); shizehuan@scdc.sh.cn (Z.S.); sujin@scdc.sh.cn (J.S.)

**Keywords:** pregnancy, dietary patterns, birth outcomes, preterm birth

## Abstract

The objective of this study was to analyse representative dietary patterns during pregnancy in Shanghai and explore the effects of dietary patterns during pregnancy on preterm birth. Data were derived from the ‘Iodine Status in Pregnancy and Offspring Health Cohort’ (ISPOHC) study. Multistage, stratified random sampling was used to select survey participants from 16 districts in Shanghai, which were divided into five sampling areas; 40–70 pregnant women were selected from each area. A total of 4361 pregnant women and their offspring were involved in the study. The male-to-female ratio of the babies was 1.04:1, and the incidence of single preterm birth was 4.2%. Three dietary patterns were extracted by factor analysis: a ‘Vegetarian Pattern’, an ‘Animal Food Pattern’ (AFP), and a ‘Dairy and Egg Pattern’. These patterns explained 40.513% of the variance in dietary intake. Binary logistic regression, which was used to analyse the association between birth outcomes and scores measuring maternal dietary patterns, found only the AFP was a significant risk factor for preterm birth. Higher AFP scores were positively associated with preterm birth (Q2 vs. Q1 OR = 1.487, 95% CI: 1.002–2.207; Q3 vs. Q1 OR = 1.885, 95% CI: 1.291–2.754). After adjusting for other potential contributors, a higher AFP score was still a significant risk factor for preterm birth (Q2 vs. Q1 OR = 1.470, 95% CI: 0.990–2.183; Q3 vs. Q1 OR = 1.899, 95% CI: 1.299–2.776). The incidence of preterm birth was 4.2%. A higher score of AFP was significantly associated with a higher risk of preterm birth. The animal food intake of pregnant women should be reasonably consumed during pregnancy.

## 1. Introduction

Preterm birth, defined as a live birth before 37 weeks of pregnancy, was the leading cause of neonatal death [1]. Preterm birth is a worldwide problem, with 15 million babies born prematurely each year, and 1 million babies dying each year due to the complications of preterm birth [2]. Even when premature babies survive, they face a greater risk of short-term and long-term morbidities, such as cognitive impairment, cerebral palsy, infections, feeding difficulties, hearing impairment, visual impairment, and neurodevelopmental impairment [3]. According to the World Health Organization (WHO), the rate of preterm birth across 184 countries ranged from 5% to 18% in 2018 [4]. Although the rate of preterm birth in China is lower than the rate in many other countries, it is rising year by year. In the past three decades, the rate of preterm birth in China has gradually increased from 5.36% in 1990–1994 to 7.04% in 2016 [5]. The rise in the incidences of preterm birth might be related to many factors, such as the increase in the proportion of births among women over 34 years of age, the increasing rates of multiple birth, the increasing choice of cesarean section and changes in diet during pregnancy [5,6]. Preterm birth not only has a profound economic effect on families and society but also increases the risk of preterm birth during a subsequent pregnancy. Therefore, the problem of preterm birth requires our urgent attention.

Pregnancy is a complex physiological process, and nutrition during pregnancy plays an important role in the health of pregnant women and the growth and development of their babies. There is substantial evidence that inadequate nutrition, such as low levels of Vitamin D, Vitamin B_12_, Vitamin E, calcium, and zinc is associated with preterm birth [7,8,9]. However, past studies have only focused on the relationship between a particular nutrient and preterm birth, whereas the association between preterm birth and diet is complex, and the synergistic and antagonistic effects of various nutrients and foods need to be taken into account [10]. Studying dietary patterns is a valuable method to examine these complex relationships between diet and disease because it more comprehensively reflects the amount of food intake, the proportions of different foods and their nutritional composition. In recent years, the analysis of dietary patterns has emerged as an important method to study the overall effects of diet. Additionally, since maternal diets, as one of the important influencing factors of preterm birth, are easier to control than other factors, we can use dietary patterns to study the relationship between maternal diets and the health of pregnant women and their babies, and thoroughly evaluate the relationship between maternal diets and preterm birth [11]. Some foreign studies have investigated the relationship between dietary patterns during pregnancy and preterm birth. Many of these have shown that the adoption of specific dietary patterns during pregnancy, such as ‘prudent diets’ [12], ‘traditional diets’ [12], a ‘DASH dietary pattern’ [13], a ‘vegetable-fruits-rice (VFR) pattern’ [14], and a ‘Mediterranean-type diet’ [15] can be beneficial for reducing the risk of preterm birth, whereas the ‘Western dietary pattern’ can increase the risk of preterm birth [16].

Unfortunately, most of the studies on the relationship between dietary patterns during pregnancy and preterm birth are from foreign countries and there are few studies from China. This is problematic because geographical and cultural differences across countries and regions contribute to differences in diet structures and eating habits of pregnant women, leading to differences in dietary patterns. As an international metropolis, Shanghai has become an intersectional point of Chinese and Western dietary cultures and enhanced them in accordance with its own dietary characteristics. Therefore, given the uniqueness of the Shanghai diet, it is worthwhile to conduct a study on the dietary patterns of pregnant women in Shanghai. This study chose data from the study of ‘Iodine Status in Pregnancy and Offspring Health Cohort’ (ISPOHC) to explore the relationship between dietary patterns during pregnancy and preterm birth in Shanghai.

## 2. Materials and Methods

### 2.1. Study Sample

This study analysed data from the ISPOHC study. This dietary survey of pregnant women was conducted in 2017 and the data on birth outcomes were collected in 2018. The study used multistage, stratified random sampling to obtain a representative sample. There are 16 districts in Shanghai, and each district was divided into five sampling areas: east, west, south, north and centre. A street was randomly selected from each sampling area, and 40–70 pregnant women were selected from each. Participants were evenly distributed in terms of the different stages of pregnancy. Only respondents with live single birth were included in the study; women who had miscarriages, stillbirth, twins, and multiple birth were excluded from the study. After a review of the literature [17] and discussions with experts, women with a daily energy intake of less than 800 kcal or more than 4000 kcal were also excluded from the study.

The Ethics Committee of the Shanghai Centre for Disease Control and Prevention approved the survey. All of the surveys were conducted after obtaining written consent from the respondents.

### 2.2. Data Collection

Participants were evenly distributed in terms of the different gestational weeks (first trimester, second trimester and third trimester). When pregnant women were confirmed to be enrolled in the study, the professionally trained investigators helped pregnant women complete a questionnaire through face-to-face interviews. The content of the questionnaire included: basic sociodemographic characteristics, behavioural factors, and the Food Frequency Questionnaire (FFQ). Food models and books containing pictures of food were used to estimate food intake. The validated and reliable FFQ [18] was used to collect data on consumption frequency and the consumption of various kinds of food by pregnant women during the past 3 months. The average daily intake of each food was calculated based on the combination of food frequency and food intake obtained from the FFQ. Similar foods were grouped together in order to extract and analyse dietary patterns. A total of 14 food groups were included in the analysis, based on the representative studies and the characteristics of the Chinese diet [14,16,19] (Table 1).

The birth information that was collected included sex, gestational age, birth weight, birth length and the head circumference of the baby. The collection of birth information was completed within 42 days after delivery.

### 2.3. Related Definitions and Classification Standards

According to the body mass index (BMI) classification standard in China [20], the pregnant women were classified as underweight (BMI < 18.5 kg/m^2^), normal weight (BMI = 18.5–23.9 kg/m^2^), overweight (BMI = 24.0–27.9 kg/m^2^), and obese (BMI ≥ 28.0 kg/m^2^). As few of the pregnant women in the study were obese before pregnancy, overweight and obese women were combined into a single group for analysis. The newborns were classified as low birth weight (LBW, birth weight < 2500 g), normal birth weight (NBW, birth weight 2500–3999 g), and macrosomia (birth weight ≥ 4000 g) [21]. With reference to the newly formulated growth reference standard for newborns with different gestational ages in China, according to the relationship between gestational age and birth weight, the newborns were classified as small for the gestational age (SGA, birth weight <10th percentile of the average weight of the same gestational age), appropriate for gestational age (AGA, birth weight in the 10th to 90th percentile of the average weight of the same gestational age), and large for gestational age (LGA, birth weight >90th percentile of the average weight of the same gestational age) [22].

### 2.4. Statistical Analysis

All statistical analyses were performed using SPSS 24. Since the variables were not normally distributed, the data are described as medians (interquartile ranges) and percentages. Categorical variables are presented as proportions and rates. Principal component analysis (PCA) was used to analyse the dietary patterns of the participants, which included 14 food groups; the maximum variance was used to extract the common factors. The KMO test and Bartlett spherical test were used to determine whether the data were suitable for factor analysis before dietary patterns were extracted. The exact number of common factors was obtained from the scree plot and eigenvalues (*λ* > 1). The composition of the absolute value of a factor load >0.4 was retained in the common factors to name the dietary patterns. After the dietary pattern was determined, a factor score was calculated for the dietary pattern corresponding to each study participant, which was the product of the average daily food intake of each food group and its weight. The dietary patterns were divided into three equal levels (from low to high: Q1, Q2 and Q3) based on the pattern scores. The binary logistic regression (forward stepwise) was used for univariate analyses. Additionally, the Q1 was utilized as the reference standard. The criterion for inclusion in the binary logistic regression model was 0.05, and the criterion for exclusion was 0.10. The results are expressed as odds ratios (ORs) and 95% confidence intervals. A *p*-value < 0.05 was considered statistically significant.

## 3. Results

### 3.1. Analysis of Dietary Pattern during Pregnancy

A total of 4361 pregnant women who met the requirements were included in the study. The results showed that the KMO = 0.839 and Bartlett spherical test was *p* < 0.001. These values were acceptable, indicating that these data were suitable for factor analysis. Among the 14 food groups, three factors were retained (Table 2). As seen in Table 2, the percent of variance explained by these three factors variance were 16.748%, 14.914% and 8.851%, respectively, and the accumulated variance explained was 40.513%. The first dietary pattern included vegetables, fruits, potatoes and their products, and cereals and their products. The second dietary pattern consisted of animal foods, such as aquatic products, poultry, and livestock. Only two food groups (dairy products and eggs) appeared in the third dietary pattern. Thus, the three dietary patterns during pregnancy can be summarised as a ‘Vegetarian Pattern’, an ‘Animal Food Pattern’ (AFP), and a ‘Dairy and Egg Pattern’.

### 3.2. General Characteristics and Birth Outcomes by Dietary Pattern

We compared differences in the general characteristics of the pregnant women and the birth outcomes of their babies across dietary patterns. The pregnant women’s pre-pregnancy body weight (*p* < 0.001, *p* = 0.011, *p* < 0.001) and pre-pregnancy BMI (*p* < 0.001, *p* = 0.012, *p* < 0.001) differed significantly across the three dietary patterns (Table 3). Pregnant height was also statistically significant in different food-score groups for the ‘Vegetarian Pattern’ (*p* = 0.034). No other statistically significant differences were found among women in the three dietary pattern groups.

As shown in Table 4, 50.9% of the babies were boys and 49.1% were girls. A total of 185 premature babies were born, representing 4.2% of all birth. The median birth length was 50.0 cm, and the median birth weight was 3339 g, with 103 babies classified as LBW (2.4%) and 268 classified as macrosomia (6.1%). The median gestational age at birth was 39 weeks; there were 405 cases (9.3%) of SGA and 739 cases (16.9%) of LGA. The median head circumference of the newborns was 34 cm. Significant differences in preterm birth and LGA were observed among those in the AFP group. No other significant differences were found.

### 3.3. The Relationship between the Scores of Maternal Dietary Patterns and Preterm Birth

Binary logistic regression, which was used to analyse the association between scores on maternal dietary patterns and preterm birth (Table 5), found higher scores on the AFP were positively associated with preterm birth (Q2 vs. Q1 OR = 1.487, 95% CI: 1.002–2.207; Q3 vs. Q1 OR = 1.885, 95% CI: 1.291–2.754). The scores for the ‘Vegetarian Pattern’ and ‘Dairy and Egg Pattern’ were not associated with preterm birth. After adjusting for educational level, annual family income, age, drinking during pregnancy, pre-pregnancy BMI, sex of the baby and the three dietary patterns, a higher score on the AFP was still a risk factor for preterm birth (Q2 vs. Q1 OR = 1.470, 95% CI: 0.990–2.183; Q3 vs. Q1 OR = 1.899, 95% CI: 1.299–2.776). Increases in AFP scores gradually increased the risk of preterm birth.

## 4. Discussion

Nutritional intake during pregnancy is important for the health of pregnant women and the growth and development of their babies. Dietary patterns have been commonly used in nutritional epidemiological studies because they describe food and nutrient intake as a whole. However, there has been little research in China, particularly in Shanghai, on the relationship between dietary patterns and birth outcomes, especially preterm birth, which are an important indicator of perinatal health in a country or region [23]. Therefore, we analysed the relationship between dietary patterns and preterm birth, using data from the ISPOHC, to provide a theoretical and practical reference for a reasonable dietary pattern during pregnancy to reduce the risk of preterm birth.

Studies on dietary patterns during pregnancy generally evaluate dietary intake using the FFQ. Compared with the 24-h recall, the FFQ has the advantages of convenience, low-cost, and accuracy with respect to an individual’s long-term average intake level [24]. Therefore, it has been widely used to study nutrition in large epidemiological cohorts. Some research also indicates that dietary patterns change little throughout pregnancy [25], so one survey at any time during pregnancy can be representative of the entire pregnancy. Moreover, a longitudinal study in a Spanish Mediterranean city reported that the dietary patterns of women before conception, at weeks 6, 10, 26 and 38 of pregnancy and at 6 months postpartum did not change as pregnancy progressed [26]. Thus, we administered the FFQ only once in this study to investigate dietary patterns during pregnancy.

In this study, 50.9% of the foetuses were male and 49.1% were female, so the male-to-female ratio was 1.04:1. There were 185 cases of preterm birth, representing an incidence rate of 4.2%. This preterm birth rate is lower than that reported by the WHO [4], which may be because only single births were included in our study. The male-to-female ratio and single preterm birth rate in our study are similar to the results obtained by other Chinese studies. A total of 24,375 live neonates in 69 hospitals in 13 cities in China, ranging in gestational age from 24 to 42 weeks, were included in the process of establishing growth standards for Chinese newborns. The male-to-female ratio of them was 1.18:1 [27]. A study by the Shanghai Centre for Women and Children’s Health that analysed the preterm birth rate of newborns in the first half of 2016 in Shanghai had a male-to-female ratio of 1.10:1, including 591 cases of preterm birth (a preterm birth rate of 5.53%), and 523 single-born preterm birth (a single preterm birth rate of 4.89%) [23]. A 2019 multi-centre study of live-birth premature infants in 53 hospitals in 17 cities in Henan Province found there were 212,438 live births (a single preterm rate of 4.4%) [28].

The current study of the diet of pregnant women in Shanghai identified three dietary patterns: a ‘Vegetarian Pattern’, an AFP, and a ‘Dairy and Egg Pattern’. The results of binary logistic regression showed that higher scores for the AFP were associated with a higher risk of preterm birth. The AFP in this study included other aquatic products, fish, poultry and livestock. This dietary pattern is characterized by a high intake of animal foods during pregnancy; such foods are rich in fat and protein, and red meat and saturated fatty acids are especially high. There are many studies abroad that have reported results similar to ours. The common denominators in terms of food between the ‘Western-type diet’ and AFP are that they both contain a mix of meat, such as pork, beef, and veal. The Danish National Birth Cohort study, which included nearly 60,000 mother-child pairs to examine the associations of dietary patterns and preterm birth, extracted seven dietary patterns, including a ‘Western-type diet’, which contained high amounts of pork, beef veal, meat mixed, meat cold, dressing sauce, potatoes and so on. The study found a high intake of this diet increased the odds of preterm birth even after adjusting for confounding factors [16]. The systematic review including 36 studies on maternal dietary patterns and preterm birth identified two basic dietary patterns: healthy and unhealthy [29]. The unhealthy dietary pattern, which was characterized by a high intake of processed meat, foods high in saturated fat or sugar and refined grains, increased the risk of preterm birth 1.17 times. Unhealthy dietary patterns were also similar to the AFP, both of which are characterized by foods high in saturated fat.

The reason why a higher score for the AFP is a risk factor for preterm birth may be related to the way diet influences inflammation. A large number of studies have shown that increased systemic inflammation and increased stimulation of infections or inflammatory pathways can all increase the risk of preterm birth [28,30]. A cohort study of a dietary inflammatory index and hs-CRP levels, which identified red meat as a pro-inflammatory food, found the intake of pro-inflammatory food was positively correlated with serum hs-CRP levels [31]. Higher maternal CRP levels have also been associated with preterm birth [32]. Red meat is one of the four main factors in the AFP, while neither of the other two dietary patterns contain red meat. A study of pro-inflammatory diets during pregnancy, which used a dietary inflammatory index to evaluate the inflammatory effect of nutrients, reported that total fats and saturated fatty acids were two of the most inflammatory nutrients, which is in line with the characteristics of the AFP [33]. On the other hand, a ‘high fat dietary pattern’ may induce obesity during pregnancy, and obesity can put the body in a state of chronic subcellular inflammation, which can also increase the risk of preterm birth [34]. Given the relationship between diet and inflammation, women should reduce the possibility of inflammation by changing their diet during pregnancy, thereby reducing the risk of preterm birth.

Omega-3 long-chain polyunsaturated fatty acid is also a hot topic in the current research on preterm birth, but there is still a controversy about whether it can reduce the incidences of preterm birth. Some studies claimed that higher intakes of foods containing EPA and DHA during pregnancy is an effective strategy for reducing the incidence of preterm birth [35,36], while the systematic review published in 2020 claimed that omega-3 fatty acid supplementation was not associated with reduced risk of preterm birth [37]. The main natural dietary source of DHA and EPA is sea fish, such as salmon, mackerel, sardines [36,38], which are regarded as low consumption foods in Shanghai. In our study, the fish accounted for only 23.65% of all animal foods, and the sea fish just 7.57%. Given the small proportion of sea fish intake in the overall animal food intake and the controversy in reducing the incidence of preterm birth, we have not singled it out for analysis.

As far as we know, this is the first community-wide study in Shanghai to investigate the effect of dietary patterns during pregnancy on offspring birth outcomes. It has the strengths of a large sample size, good coverage, and a representative sample. However, the study also has some limitations. Firstly, recall bias could not be avoided in this investigation. Secondly, diet is a multi-dimensional exposure. Although many possible confounding factors have been controlled, there are still some uncontrollable potential confounding factors, such as the environment, etc.

## 5. Conclusions

The incidence of single preterm birth in Shanghai was 4.2% in this study. This percentage is not high in comparison to other regions. However, due to the large population base and given the potential harm of preterm birth to the health of offspring, as well as the social burden, we still need to pay attention to it.

The higher score of AFP was significantly associated with higher risk of preterm birth. Therefore, it is important to rationally control the intake of animal foods during pregnancy. We should strengthen relevant nutrition and health education, and conduct more scientific research on reasonable dietary nutrition interventions during pregnancy to reduce the incidence of preterm birth, protect the health of mothers and children, and reduce the overall social burdens that come with preterm birth.

## Figures and Tables

**Table 1 nutrients-13-02367-t001:** Food groups used in the dietary pattern analysis.

Food Groups	Food Items
Cereals and their products	Cereals and their products, such as rice, oatmeal, corn, oats, noodles, steamed buns, dumplings, wontons
Potatoes and their products	Potatoes and other tubers and their products, such as vermicelli, vermicelli
Vegetables	Legume vegetables, solanaceous vegetables, cole crops, root vegetables, leaf vegetables
Beans and their products	Soybean and its products, such as soy milk, soy milk, dried bean curd
Thallophytes	Fungus, tremella, mushroom, seaweed, kelp
Fruits	Apples, pears, oranges, bananas and other fruits
Dairy products	Milk, yoghurt
Nuts	Walnut, pine nuts, almonds, hazelnuts, pistachios, peanuts and other nuts
Poultry	Chickens, ducks, geese
Livestock	Pigs, cows, sheep
Fish	Freshwater fish, sea fish
Other aquatic products	Shrimp, crabs, shells, jellyfish, squid, sea cucumbers, octopus
Eggs	Eggs, duck eggs
Salted products	Animal salted products, plant salted products, marinated meat, sauce

**Table 2 nutrients-13-02367-t002:** Factor loadings of the three dietary patterns during pregnancy.

Dietary Pattern	Factor Loadings	Eigenvalues	% of Variance Explained	% of Accumulated Variance Explained
Vegetarian Pattern		3.360	16.748	16.748
Fruits	0.617			
Potatoes and their products	0.582			
Cereals and their products	0.544			
Vegetables	0.517			
Thallophytes	0.514			
Salted products	0.489			
Beans and their products	0.477			
Nuts	0.442			
AFP		1.174	14.914	31.662
Other aquatic products	0.756			
Fish	0.741			
Poultry	0.589			
Livestock	0.557			
Dairy and Egg Pattern		1.138	8.851	40.513
Dairy products	0.754			
Eggs	0.708			

**Table 3 nutrients-13-02367-t003:** Characteristics of the participants by dietary pattern.

	*n*, %	Vegetarian Pattern			*p*	AFP			*p*	Dairy and Egg Pattern			*p*
		Q1	Q2	Q3		Q1	Q2	Q3		Q1	Q2	Q3	
Age (*n*, %)					0.736				0.775				0.528
<35 years	3771(86.5)	1262 (86.9)	1250 (85.9)	1259 (86.6)		1254 (86.2)	1264 (87.0)	1253 (86.2)		1252 (85.9)	1248 (87.3)	1271 (86.3)	
≥35 years	590(13.5)	191 (13.1)	205 (14.1)	194 (13.4)		200 (13.8)	189 (13.0)	201 (13.8)		206 (14.1)	182 (12.7)	202 (13.7)	
Educational level (*n*, %)					0.435				0.578				0.740
≤9 years	671(15.4)	212 (14.6)	238 (16.4)	221 (15.2)		213 (14.6)	229 (15.8)	229 (15.8)		229 (15.7)	218 (15.2)	224 (15.2)	
10–15 years	3182(73.0)	1057 (72.7)	1059 (72.8)	1066 (73.5)		1074 (73.9)	1042 (71.8)	1066 (73.3)		1048 (71.9)	1057 (73.9)	1077 (73.2)	
≥16 years	506(11.6)	184 (12.7)	158 (10.9)	164 (11.3)		167 (11.5)	181 (12.5)	158 (10.9)		180 (12.4)	155 (10.8)	171 (11.6)	
Family income last year, Yuan (*n*, %)					0.363				0.462				0.416
<100,000	745(17.1)	263 (18.1)	244 (16.8)	238 (16.4)		258 (17.8)	233 (16.1)	254 (17.5)		249 (17.1)	236 (16.5)	260 (17.7)	
100,000–200,000	1792(41.2)	572 (39.4)	622 (42.8)	598 (41.3)		610 (42.0)	604 (41.7)	578 (39.9)		614 (42.2)	566 (39.7)	612 (41.6)	
≥200,000	1814(41.7)	615 (42.4)	586 (40.4)	613 (42.3)		583 (40.2)	613 (42.3)	618 (42.6)		592 (40.7)	624 (43.8)	598 (40.7)	
Drinking during pregnancy (*n*, %)	51 (1.2)	11 (0.8)	23 (1.6)	17 (1.2)	0.120	19 (1.3)	12 (0.8)	20 (1.4)	0.323	17 (1.2)	17 (1.2)	17 (1.2)	0.996
Height (cm, median, P25, P75)	160.0 (158.0, 164.0)	160.0 (158.0, 164.0)	160.0 (158.0, 164.0)	161.0 (158.0, 165.0)	0.034	160.0 (158.0, 164.1)	160.0 (158.0, 164.0)	161.0 (158.0, 165.0)	0.121	160.7 (158.0, 164.5)	160.0 (158.0, 164.0)	161.0 (158.0, 164.0)	0.227
Pre-pregnancy weight (kg, median, P25, P75)	55.0 (50.0, 60.0)	55.0 (50.0, 60.0)	54.5 (50.0, 60.0)	55.0 (50.0, 60.0)	<0.001	55.0 (50.0, 60.0)	55.0 (50.0, 60.0)	54.0 (50.0, 60.0)	0.011	55.0 (50.0, 60.0)	53.0 (49.0, 58.0)	55.0 (50.0, 62.0)	<0.001
Pre-pregnancy BMI (*n*, %)					<0.001				0.012				<0.001
<18.5	540 (12.4)	172 (11.8)	200 (13.7)	168 (11.6)		174 (12.0)	201 (13.8)	165 (11.3)		171 (11.7)	219 (15.3)	150 (10.2)	
18.5–23.9	3071 (70.4)	1044 (71.9)	1041 (71.5)	986 (67.9)		1004 (69.1)	1038 (71.4)	1029 (70.8)		1043 (71.5)	1015 (71.0)	1013 (68.8)	
≥24	750 (17.2)	237 (16.3)	214 (14.7)	299(20.6)		276(19.0)	214(14.7)	260(17.9)		244(16.7)	196(13.7)	310 (21.0)	

**Table 4 nutrients-13-02367-t004:** Characteristics of birth outcomes by dietary pattern.

Characteristics	*n*, %	Vegetarian Pattern			*p*	AFP			*p*	Dairy and Egg Pattern			*p*
		Q1	Q2	Q3		Q1	Q2	Q3		Q1	Q2	Q3	
Gender (*n*, %)					0.669				0.758				0.875
boy	2134 (50.9)	714 (51.4)	722 (51.3)	698 (49.9)		727 (51.6)	699 (50.3)	708 (50.7)		713 (51.1)	703 (51.2)	718 (50.3)	
girl	2061 (49.1)	675 (48.6)	685 (48.7)	701 (50.1)		681 (48.4)	692 (49.7)	688 (49.3)		681 (48.9)	671 (48.8)	709 (49.7)	
Preterm birth (*n*, %)	185 (4.2)	62(4.3)	58(4.0)	65(4.5)	0.807	43 (3.0)	63 (4.3)	79 (5.4)	0.004	70 (4.8)	58 (4.1)	57 (3.9)	0.418
Birth weight (g, median, P25, P75)	3339 (3080, 3600)	3339 (3060, 3600)	3340 (3100, 3600)	3340 (3080, 3600)	0.665	3339 (3080, 3600)	3339 (3070, 3339)	3339 (3100, 3600)	0.457	3339 (3100, 3600)	3340 (3069, 3600)	3339 (3079, 3600)	0.767
LBW (*n*, %)	103 (2.4)	31 (2.1)	40 (2.7)	32 (2.2)	0.512	33 (2.3)	31 (2.1)	39 (2.7)	0.583	39 (2.7)	34 (2.4)	30 (2.0)	0.508
NBW (*n*, %)	3990 (91.5)	1328 (91.4)	1333 (91.6)	1329 (91.5)		1323 (91.0)	1343 (92.4)	1324 (91.1)		1322 (90.7)	1315 (92.0)	1353 (91.9)	
Macrosomia (*n*, %)	268 (6.1)	94 (6.5)	82 (5.6)	92 (6.3)	0.632	98 (6.7)	79 (5.4)	91 (6.3)	0.327	97 (6.7)	81 (5.7)	90 (6.1)	0.526
Body Length (cm, median, P25, P75)	50 (50, 50)	50 (50, 50)	50 (50, 50)	50 (50, 50)	0.355	50 (50, 60)	50 (50, 60)	50 (50, 60)	0.067	50 (50, 50)	50 (50, 50)	50 (50, 50)	0.965
Gestational age (week, median, P25, P75)	39.0 (38, 40)	39 (38, 40)	39 (38, 40)	39 (38, 40)	0.218	39 (38, 40)	39 (38, 40)	39 (38, 40)	0.301	39 (38, 40)	39 (38, 40)	39 (38, 40)	0.587
SGA (*n*, %)	405 (9.3)	149 (10.3)	127 (8.7)	129 (8.9)	0.246	132 (9.1)	139 (9.6)	134 (9.2)	0.982	146 (10.0)	126 (8.8)	133 (9.0)	0.465
AGA (*n*, %)	3217 (73.8)	1052 (72.4)	1096 (75.3)	1069 (73.6)		1048 (18.8)	1091 (75.1)	1078 (74.1)		1061 (72.8)	1055 (73.8)	1101 (74.7)	
LGA (*n*, %)	739 (16.9)	252 (17.3)	232 (15.9)	255 (17.5)	0.381	274 (18.8)	223 (15.3)	242 (16.6)	0.043	251 (17.2)	249 (17.4)	239 (16.2)	0.461
Head Circumference (cm, median, P25, P75)	34 (33, 35)	34 (33, 35)	34 (33, 35)	34 (33, 35)	0.968	34 (33, 35)	34 (33, 35)	34 (33, 35)	0.770	34 (33, 35)	34 (33, 35)	34 (33, 35)	0.237

**Table 5 nutrients-13-02367-t005:** Logistic regression results of the associations between the dietary pattern scores and preterm birth.

	Model 1 ^a^			Model 2 ^b^			Model 3 ^c^		
	*p*	OR	95% CI	*p*	OR	95% CI	*p*	OR	95% CI
Vegetarian Pattern									
Q1	Reference			Reference			Reference		
Q2	0.553	/	/	0.496	/	/	0.534	/	/
Q3	0.592	/	/	0.125	/	/	0.725	/	/
AFP									
Q1	Reference			Reference			Reference		
Q2	0.049	1.487	1.002–2.207	0.049	1.487	1.002–2.207	0.056	1.470	0.990–2.183
Q3	0.001	1.885	1.291–2.754	0.001	1.885	1.291–2.754	0.001	1.899	1.299–2.776
Dairy and Egg Pattern									
Q1	Reference			Reference			Reference		
Q2	0.670	/	/	0.558	/	/	0.611	/	/
Q3	0.383	/	/	0.216	/	/	0.216	/	/

Note: ^a^ Model 1 included the separate dietary patterns. ^b^ Model 2 adjusted for the other dietary patterns. ^c^ Model 3 = Model 2 plus educational level, annual family income, age, drinking during pregnancy, pre-pregnancy BMI, and sex of the baby.

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
