# Peer review of "Effects of Dietary Patterns during Pregnancy on Preterm Birth: A Birth Cohort Study in Shanghai"

_nutrients, 2021, doi:10.3390/nu13072367_

Round 1

Reviewer 1 Report

Dietary patterns

Overall, this study appears to have been well done and appropriately analyzed.  I have only a few comments.

Abstract: As this study design cannot prove that AFP is the cause of preterm birth (dietary patterns are correlated with many other lifestyle and other factors), the last sentence is an overinterpretation of study results.

Introduction

Is the cause for the rise in PTB known?  Is it more inductions for medical cause, or a rise in spontaneous births, or better neonatal care leading to fewer infant deaths?

The study title contains iodine levels.  Are they known to be low in this area, or is there concern about this issue?

Methods

Why call three levels Q1-Q3?

If I understand correctly, each participant received a score for each pattern, so could have scored high or low on multiple patterns.  What was the correlations among the three?

Results

Table 5.  Present the non-statistically significant as well as significant findings, and present a single p-value for the relationship between each pattern and the outcome, rather than separately for each individual level.  Also, I am not clear why Q1 (the reference category) has an associated p-value.

Discussion

The authors mention that little previous work has been done in China, which is a worthwhile criticism.  It would be useful for the authors to provide context as to how patterns that sound similar (such as “animal food pattern”) might or might not contain similar foods, foods prepared similarly or differently, or reflect cultural contexts similarly or differently (for instance, whether more affluent people are likely to eat more or fewer animal foods) than studies in other countries.

Final paragraph – same comment as the abstract; this kind of study cannot demonstrate causality so it is not known whether control of animal protein would change preterm birth or not.

Reviewer 2 Report

   It is very valuable that this paper is evaluating the association between maternal diet during pregnancy and preterm birth as dietary patterns rather than as a particular food or nutrient.

1)  As a methodology for analysing dietary patterns, 14 food groups were added to the principal component analysis in this study. The reasons for including 14 groups in the PCA should be clearly stated in this paper. In your discussion, you say that ‘the unhealthy dietary patterns containing refined grains increase the risk of preterm birth’. So why did you classify high-refined and low-refined grains together? This question cannot be answered by quoting ‘Chinese balance dietary pagoda’. I think it should be classified according to previous studies that analyze dietary patterns.

The results of analysis by PCA differ greatly depending on the classification method of food groups. It is important to indicate validity of classifying food groups.

2)  When did you conduct dietary survey during pregnancy? Please add to the paper.

3)  Table 4 shows significant differences between the ‘Animal Food Pattern’ scores and preterm birth and LGA. Based on the results, you are further analyzing the association between the maternal dietary pattern scores and the preterm birth by binary logistic regression(Table5). Did you need to do the same analysis for LGA? In addition, there is nothing in the discussion, even though there is a significant difference between the ‘Animal Food Pattern’ scores and LGA. I think you need to add the discussions on LGA.

4)  As a result of PCA analysis, the ‘Animal Food Pattern’ consisted of animal foods including fish. As you know, fish is a food rich in omega 3 fatty acids, such as EPA and DHA. Omega-3 fatty acids reduce the risk of preterm birth.

(Middleton P, Gomersall JC, Gould JF, Shepherd E, Olsen SF, Makrides M. Omega‐3 fatty acid addition during pregnancy. Cochrane Database of Systematic Reviews 2018, 11.)

Why does increases in ‘Animal Food Pattern’ scores gradually increase the risk of preterm birth, even though it contains fish? This argument is inevitable. Please consider it carefully.

5) Line 248

There is no No.87 reference in this paper, so please correct it.

Round 2

Reviewer 2 Report

The authors have modified the contents of their paper appropriately for my comments.